# Does the type of foam roller influence the recovery rate, thermal response and DOMS prevention?

Jakub Grzegorz Adamczyk[1]*, Karol Gryko[2], Dariusz Boguszewski[3]

**1** Theory of Sport Department, Józef Piłsudski University of Physical Education in Warsaw, Warsaw, Poland, **2** Athletics and Team Sport Games Department, Józef Piłsudski University of Physical Education in Warsaw, Warsaw, Poland, **3** Rehabilitation Department, Physiotherapy Division, Medical University of Warsaw, Warsaw, Poland

* jakub.adamczyk@awf.edu.pl

## Abstract

### Purpose

Supporting post-exercise recovery requires choosing not only the right treatment but also the equipment, in which the impact is not always clear. The study aimed to determine the effect of foam rolling on the rate of lactate removal and DOMS prevention and whether the type of foam roller is effective in the context of post-exercise recovery.

### Methods

This randomized trial enrolled 33 active healthy males divided into three groups of eleven individuals: foam rolling with a smooth (STH) or grid roller (GRID) or passive recovery (PAS). All the participants performed full squat jumps for one minute. Examination took place at rest (thermal imaging of skin temperature–[$T_{sk}$] and blood lactate–[LA]), immediately following exercise ($T_{sk}$ & LA), immediately after recovery treatment ($T_{sk}$) and after 30 minutes of rest ($T_{sk}$ & LA). Their pain levels were assessed using the Visual Analog Scale (VAS) 24, 48, 72, and 96 hours after exercise.

### Results

The magnitude of lactate decrease depended on the type of recovery used. In the PAS group, the decrease in lactate concentration by 2.65 mmol/L following a half-hour rest was significantly lower than that in the other groups (STH vs. PAS p = 0.042 / GRID vs. PAS p = 0.025). For thermal responses, significant differences between both experimental groups were noted only 30 minutes after exercise. A significant decrease in pain in the STH group occurred between 48 and 96 hours, while the GRID group showed a systematic significant decrease in VAS values in subsequent measurements. Changes in VAS values in subsequent measurements in the PAS group were not statistically significant (p>0.05).

**Data Availability Statement:** Data has been uploaded to figshare as one of recommended repositories. (https://figshare.com/s/1cfef1d5e1c1ca53493f).

**Funding:** The study was supported by the Polish Ministry of Science and Higher Education (Grant No. AWF – DS-273).The funders had no role in study design, data collection and analysis, decision to publish, or preparation of the manuscript.

**Competing interests:** The authors have declared that no competing interests exist.

## Conclusions

Foam rolling seems to be effective for enhancing lactate clearance and counteracting DOMS, but the type of foam roller does not seem to influence the recovery rate.

## Introduction

During intense exercise, human muscles produce and release large quantities of lactate (LA) while at the same time using it as a potential source of energy for further work [1]. Therefore, since the development and onset of fatigue correlate well with blood lactate (LA) accumulation, lactic acid measurement provides an effective and accessible measure that can contribute to fatigue [2]. Moreover, most effective methods of lactate clearance during recovery and their impact on performance have attracted the attention of sports scientists and practitioners. Despite many post-exercise recovery treatments, only a few are considered effective. Some examples of the methods used for improving recovery include stretching, massage, ice massage and cold water immersion, transcutaneous muscle stimulation, kinesiotaping, music therapy, low-intensity exercise, and recovery using pharmacological agents [3, 4, 5]. Most of them are based on improving blood flow, which can be done by different methods of stimulation (e.g., temperature, mechanical impact, muscle movements).

Increasing blood flow through muscles following exercise can help quickly eliminate fatigue symptoms. Therefore, low-intensity exercise can have a positive effect on post-exercise muscle recovery [6]. Recovery can also be supported by mechanically increasing the compression on tissues to induce vasodilation and increase blood circulation [4]. One of the methods of achieving this is self-myofascial release (SMFR) by means of a foam roller, which can increase the hydration and elasticity of fascia. Foam rolling involves using the pressure of a person's own body weight on a foam roller and therefore on soft tissue during movement [7].

The effect of increased arterial blood flow may induce the physiological mechanisms preventing muscle fatigue associated with physical exercise. Foam rolling treatment is associated with improved arterial perfusion by increasing blood flow in the arteries, so it seems that foam rolling (FR) may lead to physiological adaptive changes in increased efficiency, ROM and recovery [8, 9, 10]. Studies in the field of SMFR indicate the effectiveness of this technique in relieving pain due to many physiological responses. Some of them are increased blood flow, reduced arterial stiffness, improved vascular endothelial function and increased nitric oxide concentration [11].

Research also suggests that the sensation of DOMS (delayed onset muscle soreness) may be weaker when using the foam rolling technique due to thixotropy, which could cause locally altered tissue stiffness or nonneural tone [12]. Such an approach promotes the gel-like state of fascia without any impairment of neuromuscular properties [13]. In a study by Bradbury-Squires et al. [14], these symptoms were significantly reduced in each subsequent test (24 h, 48 h and 72 h) in persons who performed FR once a day with two sets of 60 seconds each. Delayed onset muscle soreness is primarily caused by changes in connective tissues, and foam rolling affects mainly connective tissue and not muscle tissue. This explains the reduction in pain sensation without a visible loss of muscle function [15]. Another underlying cause is the increase in blood flow, which leads to the removal of blood lactic acid, reduction in swelling and oxygen supply to the muscles [9].

D'Amico and Paolone [16] studied the effect of rolling on the recovery between two anaerobic efforts and found that FR may not be an effective way to support recovery between

intensive exercises with a 30-minute rest. According to the authors, foam rolling during recovery after exercise-induced muscle damage caused by sprinting does not seem to be effective for reducing muscle pain beyond what can be achieved with a dynamic warm-up. Hodgson et al. [17] also questioned the long-term effects of foam rolling on pain sensation as immediate effects of foam rolling may not translate into chronic changes probably due to central pain modulation which might require longer SMFR influence.

The factor potentially affecting the effectiveness of the SMFR procedure is the type of foam roller used for exercises. Greater hardness and a nonuniform structure (grid) contribute to increasing the point pressure on the massaged tissue so that the analgesic effect may be greater [18, 19]. In addition to increased blood circulation, SMFR is associated with trigger-point release, which, with the use of grid foam rollers, might increase pressure on tissue and enhance recovery [20].

Safe, noninvasive diagnostic methods are needed to ensure proper training processes and health care in athletes. An example of such methods is thermal imaging. Information about the efficiency of metabolic changes and endogenous heat removal systems during training associated with the return to homeostasis and post-exercise recovery is indicated by the change in body surface temperature. Thermal imaging techniques offer opportunities for monitoring these phenomena. The findings published by Adamczyk et al. [21] showed that maximum anaerobic exercise was accompanied by a significant decline in temperature on the surface of the involved muscles, whereas the degree of reduction was proportional to the blood lactate concentration. Lactic acid levels effectively reflect muscle fatigue during and after exercise [1]. Lactate measurement can be performed immediately following or even during exercise because it does not require complex laboratory procedures [22]. Thermal imaging allows for a psychologically comfortable evaluation of post-exercise recovery without substantial financial outlays [23].

Heterogeneity in research makes difficult to identify a consensus on an optimal SMFR program and the optimal use of FR [7]. Several important issues related to foam rolling, such as optimal duration [24], foam roller density and tissue pressure [25], remain unexplained. Differences in the force acting on muscles during foam rolling caused by body weight and individual differences in techniques can lead to different effects of foam rolling [26]. Taking this into account, the aim of the study was to determine whether the type of roller in single foam rolling treatment influences the rate of lactate removal and DOMS prevention.

## Materials and methods

### Participants

The randomized trial enrolled 33 active healthy untrained male participants randomly divided into three groups (Table 1) of eleven people depending on the type of recovery: rolling with a smooth (STH) or grid (GRID) foam roller or passive recovery (PAS). Their mean age was 24.5 years (±2.9), mean body height was 182.0 cm (±5.7), and mean body mass was 82.7 kg (±9.4).

**Table 1. Biometric characteristics of the participants divided into groups, mean values (±SD).**

| Group | SMOOTH ROLLER (STH) n = 11 | GRID ROLLER (GRID) n = 11 | PASSIVE REST (PAS) n = 11 |
|---|---|---|---|
| Age [years] | 24.4 ±3.4 | 24.5 ±2.9 | 24.1 ±4.4 |
| Body height [cm] | 182.8 ±5.3 | 182.0 ±5.7 | 181.5 ±6.6 |
| Body mass [kg] | 82.3 ±5.4 | 82.7 ±9.4 | 83.2 ±7.2 |

All participants had a normal body mass index (BMI). The study was conducted in October 2019. Each group performed test separately but experimental sessions for groups, were held at the same time of the day. Participants were informed about the risks and provided their written informed consent. The study was approved by the Research Ethics Committee of the Józef Piłsudski University of Physical Education in Warsaw (No. SKE 01-41/2016).

## Experimental approach to the problem

Before the experiment, the participants were rest subjected to 20 minutes of thermal adaptation to the conditions of the room where the examinations were performed. The purpose of the adaptation was to achieve a state of thermal balance in relation to the ambient temperature so that the obtained thermograms were reliable. Therefore, changes in the thermal images were the result of disturbances in production or heat dissipation as a result of the exercises [27]. Resting blood lactate levels from a capillary earlobe sample were measured and analyzed with a Dr. Lange LP 420 photometer (Germany) [28]. After the adaptation, a thermal image of the lower limbs was taken. Due to adaptation before thermal imaging, all participants wore shorts to expose as much of the lower limbs as possible.

After the completion of adaptation and thermal images at rest conditions (REST), the participants performed the exercise test described in our previous studies [21, 29]. The test consisted of maximum-effort squat jumps performed for one minute. The test was developed to be performed in a short time with limited space and allowed for inducing a significant increase in blood lactate levels as an effect of the glycolytic nature of exercise [21, 29]. During our preliminary research, creatine kinase (CK) activity was also measured 24 hours following the same exercise. The average CK values ranged between 300 and 500 U-L-1, which strongly suggests high-intensity exercise and the possible occurrence of DOMS [30].

Immediately after the trial (IAT), the second round of thermal imaging and blood lactate measurement was carried out. Next, the lower limbs were foam rolled in the STH and GRID groups. A portion of the respondents used smooth foam rollers of medium density, size 30 cm x 15 cm, while the other group performed rolling using grid foam rollers of medium density with the same size. The participants performed a cycle of 30 complete movements with a frequency of 50 beats per minute [18]. Bodyweight was used for each muscle group, starting with the gastrocnemius muscle, followed by the hamstring muscles, quadriceps muscle, adductor group, iliotibial band and gluteus muscle. Both lower limbs were subjected to this procedure. The PAS group was instructed to rest passively.

After foam rolling treatment (ART), each participant had a thermal image taken and then rested until 30 minutes (AFTER30) from the completion of the exercise. Again, a thermal image was taken, and blood lactate measurement was performed. Furthermore, 24, 48, 72 and 96 hours after the completion of the activity, the participants were asked to assess their lower extremities pain sensation by completing the VAS (Visual Analog Scale) from 0 to 10, in which 0 indicated "no pain" and 10 represented "extreme pain" [31, 32]. Only subjects who declared no pain symptoms, both in terms of the lower limbs and general body pain, before the examination were included in the study.

Thermograms of the front and back surfaces of the lower limbs for each participant were taken in a standing position to determine skin temperature ($T_{sk}$). The analyzed area was divided into the following regions of interest (ROIs): anterior thigh (from the deflection in the hip joint to the knee—excluding the patella), anterior calf (from the tibial tuberosity to the ankle), posterior thigh (from the gluteal folds to the knee—excluding the popliteal fossa) and posterior calf (from the bottom of the popliteal fossa to the ankle). A thermal imaging camera (FLIR A325, FLIR Systems, Sweden) was used for all infrared (IR) measurements. The camera

had a measurement range from -20 to +350˚C, an accuracy of ±2˚C or ±2%, a sensitivity below 0.05˚C, an infrared spectral band of 7.5–13 μm, a refresh rate of 60 Hz, and a resolution of 320–240 pixels of FPA. The distance between the camera and the photographed object was set at 2.5 m. Recommendations for thermal imaging in sports and exercise medicine were followed as described in previous research [33]. The analysis was performed with the use of Researcher 2.9 Pro software designed for use with the thermal camera.

## Statistical analysis

For the sample size (3 groups, each n = 11), assuming a typical significance level alpha = 0.05 (two-way test) and the standarized effect RMSSE = 0.77, the test power was 0.88. The assumption of compliance of the distributions of the variables with the normal distribution was tested using the Shapiro-Wilk test. The assumption of the equality of variance in groups was evaluated using Levene's test. In the repeated measures procedure, the assumption of sphericity of variance was also verified (Mauchley test). To determine the significance of differences in values (LA, VAS, $T_{sk}$) in subsequent measurements, we used ANOVA with repeated measures (Bonferroni post hoc test). ANOVA for factorial designs (Bonferroni post hoc test) was used to determine the differences in values between each condition (STH, GRID and PAS). Effect size measures used the eta-squared (η2) statistics: small effect, <0.10; medium effect, 0.10–0.40; and large effect, >0.40 [34]. The relationships between the magnitude of lactate decline, VAS indications and lower limb temperature were analyzed using the Pearson correlation coefficient. The level of $p < 0.05$ was set in all analyses to assess the significance of the effects. All the statistical analyses were performed using STATISTICA software (v. 13, Stat. Soft. USA).

## Results

### Lactate

No significant differences between the groups were found in any of the measurements. The resting lactate (LA) level and the changes following the exercise test and recovery were similar in all groups (Fig 1) ($F_{(4,60)}$ = 1.61; p = 0.18; η2 = 0.10). Immediately after the trial (IAT), a significant ($F_{(1,32)}$ = 293.9; $p < 0.001$; η2 = 0.90) increase in the blood lactate level was observed, which demonstrates the glycolytic anaerobic nature of exercise. Thirty minutes of recovery was enough to significantly reduce blood lactate levels ($F_{(1,32)}$ = 92.9; $p < 0.001$; η2 = 0.74), however LA levels remained higher than at the resting state. The magnitude of lactate decline depended on the form of recovery used ($F_{(2,30)}$ = 3.38; p = 0.04; η2 = 0.18). The highest decrease in lactate (Δ 4.94 mmol/L) was observed in the GRID group. However, this difference was not significant (p = 0.82) compared to the STH group (Δ 4.25 mmol/L). The decrease in lactate concentration by 2.65 mmol/L following a half-hour rest was significantly lower than that in the other groups (STH vs. PAS p = 0.042 / GRID vs. PAS p = 0.025).

### Temperature changes

As expected, exercise caused a significant drop in skin temperature ($T_{sk}$) in most of the regions of interest (ROIs). Evaluation of the acute effect of foam rolling on skin temperature revealed no significant changes between the 2nd and 3rd measurements (p > 0.05). The greatest diversity in thermal responses was observed for the 4th measurement. In the STH group, the gradually increasing temperature reached the resting state level. At the same time, the GRID group had a significantly higher $T_{sk}$ in the anterior thigh, while the posterior thigh reached a temperature close to the resting state level (Table 2).

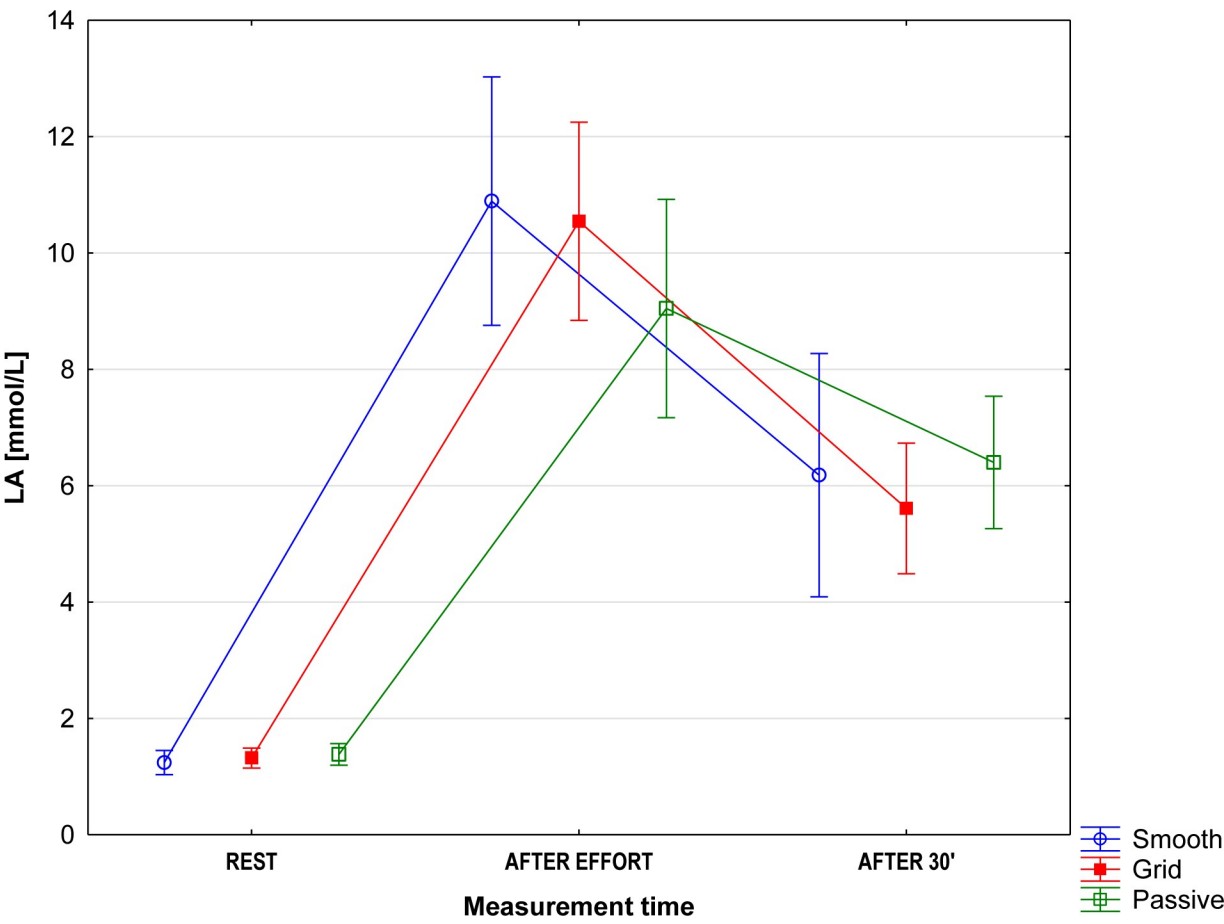

**Fig 1. Changes in blood lactate levels, mean values (95% CI).**

After initial acclimation, the temperatures of the lower limbs in all groups were similar to each other. The only significant difference was noticed for the posterior calf ($F_{(2,30)} = 4.4$; $p = 0.02$; $\eta2 = 0.23$) between the PAS and GRID groups ($p = 0.03$). No relevant differences were found in the second measurement. Furthermore, in one ROI (posterior calf), a difference occurred after foam rolling ($F_{(2,30)} = 3.7$; $p = 0.04$; $\eta2 = 0.20$), but this was between the PAS and STH groups ($p = 0.03$). The largest difference between groups was observed after 30 minutes of recovery. The only ROI with no significant differences between groups was the posterior thigh. Relevant changes were observed ($F_{(2,30)} = 3.91$; $p = 0.03$; $\eta2 = 0.21$) between the PAS and GRID groups ($p = 0.049$) for the anterior thigh. Furthermore, in the anterior calf, significant differences ($F_{(2,30)} = 4.51$; $p = 0.02$; $\eta2 = 0.23$) were found for the STH and PAS groups ($p = 0.02$). In the posterior calf, temperatures differentiated ($F_{(2,30)} = 7.58$; $p = 0.002$; $\eta2 = 0.24$) the PAS group from both the STH ($p = 0.002$) and GRID ($p = 0.04$) groups. No significant correlations between the lower limb temperature and both the LA level and $\Delta$LA were found in the thirtieth minute, regardless of the group ($p > 0.05$).

## Visual Analog Scale

Pain was significantly reduced in all groups 96 hours after completing exercise. The evaluation of pain sensation between 24 and 72 hours after the completion of exercise did not reveal any significant differences between the groups. Due to the type of support of the post-exercise recovery,

**Table 2. Mean values [˚C] of the anterior and posterior skin surface temperature of ROIs in consecutive measurements.**

| | | Measurement 1 Resting state (REST) | | Measurement 2 Immediately After Trial (IAT) | | Measurement 3 After foam rolling treatment (ART) | | Measurement 4 30th minute of recovery (AFTER30) | |
|---|---|---|---|---|---|---|---|---|---|
| | | Mean±SD | CI 95% | Mean±SD | CI 95% | Mean±SD | CI 95% | Mean±SD | CI 95% |
| Smooth | Anterior thigh | 29.95±1.12 *(2) | 29.20–30.7 | 28.7±0.90 ***(4)*(1) | 28.09–29.3 | 29.19±1.55 *(4) | 28.15–30.24 | 30.48±1.44 ***(2)*(3) | 29.51–31.45 |
| | Posterior thigh | 30.02±1.59 **(3)*(2) | 28.96–31.09 | 28.87±0.98 *(1,4) | 28.21–29.53 | 28.37±1.56 ***(4)**(1) | 27.33–29.42 | 30.16±1.02 ***(3)*(2) | 29.47–30.84 |
| | Anterior calf | 29.68±1.31 ***(2)**(3) | 28.8–30.56 | 28.17±1.52 ***(1,4) | 27.15–29.19 | 28.31±0.88 **(1.4) | 27.72–28.9 | 29.66±0.92 ***(2)**(3) | 29.04–30.28 |
| | Posterior calf | 29.98±1.24 **(2,3) | 29.15–30.81 | 28.54±1.37 **(1,4) | 27.61–29.46 | 28.6±0.70 **(1,4) | 28.12–29.07 | 29.92±0.65 **(2,3) | 29.48–30.36 |
| Grid | Anterior thigh | 29.36±1.93 **(4) | 28.07–30.65 | 28.43±1.56 ***(4) | 27.39–29.48 | 28.78±1.60 ***(4) | 27.71–29.86 | 30.51±1.33 ***(2,3)*(1) | 29.62–31.4 |
| | Posterior thigh | 29.83±2.35 *(3) | 28.25–31.41 | 28.49±1.66 | 27.38–29.61 | 28.32±1.47 *(1) | 27.33–29.31 | 29.77±1.66 | 28.66–30.89 |
| | Anterior calf | 29.63±1.48 ***(3)*(2) | 28.63–30.62 | 28.65±1.30 *(1) | 27.78–29.52 | 28.15±1.43 ***(1)**(4) | 27.19–29.11 | 29.29±1.00 **(3) | 28.62–29.96 |
| | Posterior calf | 30.15±1.14 ***(2,3) | 29.39–30.92 | 28.67±1.08 ***(1) | 27.95–29.4 | 28.16±1.11 ***(1)**(4) | 27.42–28.91 | 29.49±1.26 **(3) | 28.65–30.34 |
| Passive | Anterior thigh | 28.76±0.31 **(2) | 28.55–28.96 | 27.8±0.78 ***(4)**(1) | 27.27–28.32 | 28.34±0.93 **(4) | 27.71–28.96 | 29.27±0.63 ***(2)**(3) | 28.84–29.69 |
| | Posterior thigh | 28.76±0.44 **(2,4) | 28.46–29.05 | 27.86±0.66 ***(4)**(1) | 27.42–28.31 | 28.3±0.91 ***(4) | 27.69–28.91 | 29.62±0.66 ***(2,3)**(1) | 29.18–30.06 |
| | Anterior calf | 28.65±1.28 *(3) | 27.78–29.51 | 27.76±0.90 | 27.16–28.37 | 27.57±1.04 *(1,4) | 26.87–28.26 | 28.55±0.70 *(3) | 28.08–29.02 |
| | Posterior calf | 28.95±0.59 ***(2,3) | 28.56–29.34 | 27.78±0.78 ***(1)*(4) | 27.25–28.3 | 27.54±0.89 ***(1)**(4) | 26.95–28.14 | 28.51±0.51 **(3)*(2) | 28.16–28.85 |

* p<0.05

** p<0.01

***p<0.001 significant difference vs REST (1) / IAT (2) / ART (3) / AFTER30 (4)

significantly lower VAS values were observed only 96 hours after the study (Fig 2). Analysis of the decreased pain sensation in the STH group showed significant changes between 48 and 72 hours (p = 0.018) and 72 and 96 hours (p = 0.020). A gradual significant decrease in VAS was found in the GRID group in subsequent measurements (24/48 h p = 0.037; 48/72 h p = 0.023; 72/96 h p = 0.001). Changes in VAS in subsequent measurements in the PAS group were not statistically significant (p>0.05). No statistically significant correlations were demonstrated between the degree of lactate decrease and the indications of VAS (p>0.05) and $T_{sk}$ (p>0.05).

## Discussion

The aim of this study was to determine the effect of foam rolling on the rate of lactate removal and DOMS prevention and whether the type of foam roller was effective in the context of post-exercise recovery. The findings suggest that using a foam roller as a self-myofascial release technique is an effective approach in supporting post-exercise lactate removal, but does not prevent the pain associated with damaged muscle fibers (DOMS). Furthermore, the type of foam roller does not seem to increase the recovery rate.

### Blood lactate changes

The participants were subjected to maximum anaerobic exercise. However, no significant differences were found in the magnitude of the decrease in LA 30 minutes after the exercise, but

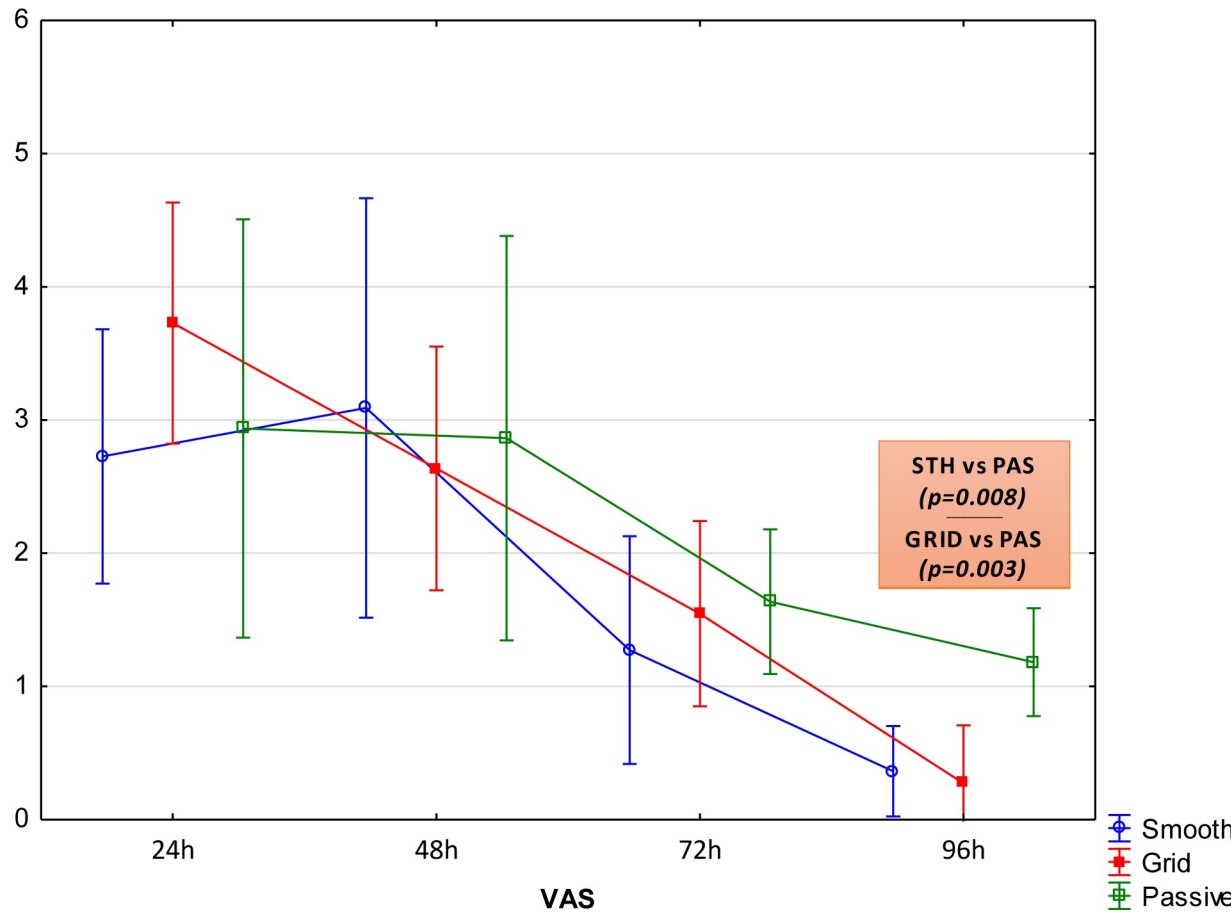

**Fig 2. VAS values for the individual groups examined in subsequent measurements, mean values (95% CI).**

both groups using the methods supporting recovery had a higher decrease in lactate than the group that rested passively. This is consistent with research on the predominance of active forms of recovery over passive ones. Some authors suggest that with the use of active methods to support recovery, blood flow in muscles is improved and lactate is removed more quickly [6]. Studies have shown that foam rolling also affects the cardiovascular system, reduces arterial stiffness, improves vascular endothelial function and increases blood flow [11, 35].

To the best of our knowledge, no unequivocal results of research on the effect of foam rolling on blood lactate levels after intense exercise have been published to date. Shalfawi et al. found that using foam rolling as a self-myofascial release technique during recovery did not increase the rate of blood lactate clearance in the cycling test to exhaustion followed by a Wingate performance test [36]. Moreover, Moraska et al. [37] stated that single trigger point release massage affected anaerobic metabolism. Our research did not support this theory, and the difference may be due to the size of the massaged muscle surface as well as the higher tolerance to high-intensity exercise and the recovery capacity of subjects.

The use of foam rolling is a form of self-massage through mechanical pressure on tissues. The potential benefits and mechanisms of massage are similar to foam rolling, including increased blood circulation and venous return, greater lactate clearance and decreased pain sensation [38, 39]. The results of this form of supportive recovery are also inconclusive. Some studies failed to show any effect of massage on improving lactate clearance. Bielik [40]

observed that massage did not reduce the blood lactate concentration after intensive physical activity and that active methods (associated with low-intensity muscle work) of recovery were more effective in reducing lactate levels. A possible explanation was presented by Hinds et al. [41], who demonstrated that massage increased skin blood flow but without an increase in arterial blood flow, which seems to be crucial for lactate clearance. On the other hand, Özsu et al. [42] showed better quality of recovery after using the self-myofascial release technique. Furthermore, the findings published by Adamczyk et al. [29] showed a significantly higher decrease in lactic acid after ice massage compared to the control group. A potential explanation may be that in contrast to massage, foam rolling requires low-intensity muscle activity. Therefore, seems that active methods to support recovery are more effective than massage in removing LA following exercise, while massage is more effective than passive rest [43]. Another explanation might be the set frequency of treatment, as self-regulated intensities showed enhanced lactate clearance [44].

## Temperature changes

Thermal imaging was carried out to assess changes in temperature, which is closely related to the recovery and evaluation of DOMS following intensive exercise [45]. Thermography has been reported in the literature as an effective technique to diagnose post-exercise pain [21, 29, 45], especially during the first 24 hours after physical activity. Furthermore, Adamczyk et al. [21] documented correlations between the decrease in body surface temperature in working muscles after maximum anaerobic exercise and the degree of decrease in blood lactate levels.

As mentioned above [21], the study did not show any significant differences in blood lactate levels, but the magnitude of lactate decline 30 minutes following exercise was different. Skin temperature ($T_{sk}$) showed significant differentiation. The test results showed a significant decrease in the surface temperature of the lower limbs of approximately 1.2–1.5˚C immediately after exercise (Table 2), which is in line with previous results [21, 29]. Reduced skin surface temperature is a typical thermoregulatory response to exercise due to redirected blood flow. These changes depend on exercise, and the more intensive the physical activity is, the greater the changes in body surface temperature [29]. It is important to note that there were absolutely no significant changes in skin temperature between immediately after exercise and after foam rolling treatment (28.31˚C and 28.30˚C, respectively). This contradicts the findings on the influence of massage on thermal response, where skin temperature was elevated after the application of massage [41]. The activation of recovery processes was confirmed by at least a 1˚C (Table 2) temperature increase after 30 minutes from the completion of exercise in each group. This occurs as a result of the return of peripheral blood to the circulation typical of resting conditions and the increased activation of involved muscles that release more energy after being exercised [21]. This strongly suggests a delayed effect of foam rolling for enhancing recovery throughout the redirection of blood flow to recovering muscles; in particular, the cutaneous vascular response to intensive effort consists of temporarily reducing blood flow in the skin, narrowing the vessels, and dilating the blood vessels to remove excess heat [46]. Al-Nakhli et al. [45] also found that increased skin temperature following exercise may result from increased blood flow in muscles due to inflammation and recovery of damaged tissues. The analysis of temperature changes reveals that they are more varied on the shin surface, most likely due to lower muscle mass and thus increased pressure.

Murray et al. [47] studied the effect of foam rolling on muscle temperature and flexibility, but contrary to the above results, they did not find any significant temperature change. Significant differences between both experimental groups were documented only 30 minutes after exercise. This may demonstrate an increased metabolic activity caused by foam rolling, which

seems to be conducive to recovery. This theory is supported by both the observed higher values of $\Delta T_{sk}$ and $\Delta VAS$ in groups using foam rolling techniques. The potentially divergent results may have been caused by the different types of foam rollers used as the pressure force, and the stimulus penetration depth of myofascial release is different [25]. Additionally, it is difficult to determine the exact pressure and duration required for each muscle during foam rolling, as the thermal response of the muscles depends on the applied pressure during treatment [48].

No differentiation in thermal response due to the type of foam roller was demonstrated. During foam rolling, the trigger point may escape from the roller's pressure, which may reduce the effectiveness of the therapy and increase the tension response of the athlete by pressing the trigger point as a result of the response to blood flow restriction in the trigger point area. This reaction restores normal blood and lymph circulation. At the same time, the process of removal of metabolites within the compressed tissue can be intensified. This improves circulation and reduces tissue adhesion [49, 50]. The effectiveness of this method was confirmed for foam rolling, but the type of foam roller was not important.

## DOMS

The values of VAS observed in our study showed a typical timeline of pain sensations [51]. Pain complaints rated throughout the VAS are expected to peak between 24 and 48 hours post-exercise [32], but it must be noted that the VAS and pressure pain threshold are associated with the different ways to quantify pain sensation. However, grid foam rollers are able to apply significantly higher pressure on the soft tissue than smooth rollers during treatment [25], and the positive effect of foam rolling on pain sensation was revealed only 96 hours after exercise. Possible that differences should be revealed earlier, but the analysis of VAS changes over time ($\Delta VAS$) suggests the effectiveness of foam rolling compared to passive rest. This result is partly reflected in the analysis of the literature. Beier et al. [52] observed that FR does not produce measurable benefits in terms of the sensation of fatigue after 24 hours. On the other hand, Drinkwater et al. [53] emphasized that foam rolling appears to improve performance in the later stages of recovery following eccentric exercise, which might explain the positive effects observed after 96 hours. Foam rolling was also beneficial for minimizing muscle soreness in research by McDonald et al. [54]. In light of research, compression of the tissues located above a cylindrical roller increases the range of motion (ROM) while maintaining power and strength [55] and brings relief by relaxing muscle tension and restoring muscle flexibility [56]. Furthermore, D'Amico and Paolone [16] observed the best effects of recovery in a group using FR compared to those who rested passively. Analysis of the VAS in the study revealed a better mood in people who performed rolling with both smooth and grid foam rollers, but there were no differences in the speed or quality of recovery due to the type of foam roller. This would undermine a fairly common theory [18] of deeper tissue penetration achieved by means of grid foam rollers.

According to our study, the effectiveness of foam rolling treatment for enhancing post-exercise recovery has been confirmed, but we found no effect of foam roller type on the recovery quality. However, there are some limitations of this analysis. First, our study included rather small groups, which might have resulted in reduced statistical power of the provided analysis. Second, exercise was limited to a single foam rolling procedure, so the influence of treatment time, frequency and roller density still remains unclear. Consequently, there may be some differences in the pressure exerted on muscle tissue through FR. These discrepancies may explain some of the inconsistent findings within the current literature, and future research should focus on finding an optimal foam rolling program.

## Conclusions

The study results confirm the effectiveness of foam rolling in the support of recovery. Foam rolling might be implemented after workouts to enhance recovery between training sessions [57]. There is also evidence of a favorable circulatory response after foam rolling, a reduction in arterial stiffness, an improvement in vascular endothelial function and an increase in arterial blood flow [37]. The use of foam rolling after training might be effective in some cases (e.g., to increase sprinting performance and flexibility or to reduce muscle pain sensation) [58]. As we observed some substantial effects of foam rolling as an effective DOMS recovery modality, practical application for accelerated recovery after exercise might be crucial for athletic training or competing with short durations of rest [18]. Based on the examinations performed in the present study, the type of foam roller is irrelevant for the effects of foam rolling.

## Author Contributions

**Conceptualization:** Jakub Grzegorz Adamczyk.

**Data curation:** Jakub Grzegorz Adamczyk, Karol Gryko.

**Formal analysis:** Jakub Grzegorz Adamczyk, Karol Gryko, Dariusz Boguszewski.

**Investigation:** Jakub Grzegorz Adamczyk, Dariusz Boguszewski.

**Methodology:** Jakub Grzegorz Adamczyk, Karol Gryko, Dariusz Boguszewski.

**Project administration:** Jakub Grzegorz Adamczyk.

**Supervision:** Jakub Grzegorz Adamczyk.

**Writing – original draft:** Jakub Grzegorz Adamczyk.

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
