## [Decision Letter · Decision Letter 0]

3 Apr 2020

PONE-D-20-05356

Does the roller type influence on recovery rate, thermal response and DOMS prevention?

PLOS ONE

Dear Mr Adamczyk,

Thank you for submitting your manuscript to PLOS ONE. After careful consideration, we feel that it has merit but does not fully meet PLOS ONE’s publication criteria as it currently stands. Therefore, we invite you to submit a revised version of the manuscript that addresses the points raised during the review process.

ACADEMIC EDITOR: In particular, please carefully proof read the manuscript for grammatical errors and typos to improve the clarity of the manuscript

We would appreciate receiving your revised manuscript by May 18 2020 11:59PM. To enhance the reproducibility of your results, we recommend that if applicable you deposit your laboratory protocols in protocols.io, where a protocol can be assigned its own identifier (DOI) such that it can be cited independently in the future. For instructions see: http://journals.plos.org/plosone/s/submission-guidelines#loc-laboratory-protocols

We look forward to receiving your revised manuscript.

Kind regards,

Kelly Naugle, PhD

Academic Editor

PLOS ONE

2. Thank you for stating the following financial disclosure: "NO"

3. Thank you for stating the following in your Competing Interests section:  "NO"

4. Please upload a new copy of Figure 1 as the detail is not clear. Please follow the link for more information: http://blogs.PLOS.org/everyone/2011/05/10/how-to-check-your-manuscript-image-quality-in-editorial-manager/

Reviewers' comments:

Reviewer's Responses to Questions

**Comments to the Author**

1. Is the manuscript technically sound, and do the data support the conclusions?

Reviewer #1: Yes

Reviewer #2: Yes

2. Has the statistical analysis been performed appropriately and rigorously? 

Reviewer #1: Yes

Reviewer #2: Yes

3. Have the authors made all data underlying the findings in their manuscript fully available?

Reviewer #1: Yes

Reviewer #2: Yes

4. Is the manuscript presented in an intelligible fashion and written in standard English?

Reviewer #1: Yes

Reviewer #2: No

5. Review Comments to the Author

Reviewer #1: The current manuscript looks to determine if there is a difference between foam roller type versus passive recovery following a bout of maximum effort jump squats. Thermal imaging was used to measure changes in blood flow via skin temperature and blood lactate was measured to gauge the effectiveness of the exercise and determine difference in immediate recovery. It was determined that the type of foam roller doesn’t affect the clearance of lactate or symptoms associated with DOMS, however, it was found to show some improvements versus passive recovery. While the manuscript is well written, I do believe it needs revisions before it can be published.

Reviewer #2: see uploaded document from the reviewer as an uploaded word document.

6. PLOS authors have the option to publish the peer review history of their article (what does this mean?). If published, this will include your full peer review and any attached files.

Reviewer #1: No

Reviewer #2: No

---

## [Author Response · Author response to Decision Letter 0]

2 May 2020

REVIEWER #1

General 

The current manuscript looks to determine if there is a difference between foam roller type versus passive recovery following a bout of maximum effort jump squats. Thermal imaging was used to measure changes in blood flow via skin temperature and blood lactate was measured to gauge the effectiveness of the exercise and determine difference in immediate recovery. It was determined that the type of foam roller doesn’t affect the clearance of lactate or symptoms associated with DOMS, however, it was found to show some improvements versus passive recovery. While the manuscript is well written, I do believe it needs revisions before it can be published. 

COMMENTS 

Abstract 

Page 2, Lines 27-28: The first sentence should start out “The study…” 

Page 2, Line 29: Males should be plural. 

Page 2, Line 32: you should have [] around Tsk and LA to indicate abbreviations. 

AUTHORS' RESPONSE: Included in the text.

Introduction 

General 

You switch back and forth between using the term roller vs. foam roller. You should pick one and make it consistent throughout. 

AUTHORS' RESPONSE: This has been adressed – now we’re using foam roller throughout all manuscript.

Page 2, Line 50: It’s the first time you use the abbreviation of LA in the main body, so you need to write it out the first time, followed by the abbreviation. 

AUTHORS' RESPONSE: Included in the text.

Page 3, Line 52: You use the provider “Physiotherapist”. However, many other healthcare professionals use foam rollers for treating athletes, so I would suggest making it more encompassing of all professions. 

AUTHORS' RESPONSE: This part has been removed in order not to limit potential users.

Page 3, Lines 63-64: Your 1st sentence is confusing. What is it compared too? Not sure this sentence should be your lead. Your paragraph is talking about arterial flow, so maybe swap 

AUTHORS' RESPONSE: This part has been rebuilded with suggested swaping between the 1st and 2nd sentences.

Page 3, Line 67: Remove the ‘a’ from the sentence. 

AUTHORS' RESPONSE: Included in the text.

Page 3, Line 70: You need to write out DOMS the first time you use it. Don’t assume everyone knows what DOMS stands for.

AUTHORS' RESPONSE: Included in the text.

Page 4, Lines 105-107: I would include a statement about the foam rolling being a single bout so not to confuse since measurements are happening up to 96 hours. 

AUTHORS' RESPONSE: Included in the text. Page 5, lines 117-118.

Methods 

Page 4, Line 127-128: The first sentence needs a reference. 

AUTHORS' RESPONSE: Included in the text.

Page 6, Line 134: You can delete IAT since you never use the abbreviation again.

AUTHORS' RESPONSE: It is used again in Results section both in the text and Tab. 2. 

Page 6, Line 142: Same as above. ART is only used here. 

AUTHORS' RESPONSE: It is used again in Tab. 2.

Page 6, Line 142: I believe you have a typo stating 20 minutes. Otherwise this sentence is very confusing. 

AUTHORS' RESPONSE: This sentence has been improved.

Page 6, Lines 148-151: On your ROI you define thigh, both anterior and posterior, as going down to the ankle. Thigh is a Latin term meaning femur, so you need to change your terminology. Plus your two tight measurements include the two calf measures.

AUTHORS' RESPONSE: I'm not sure if I understand correctly your comment on femur, as thigh refers to „upper leg”. Not the femur exclusively. Of course it’s our fault with ankle and it should be one description for thigh and second for calf. This has been clarified. Page 7, lines 161-166.

Results 

General 

The figures in your results should show were any significant differences occur. They should stand-alone and a reader should be able to look at just the figures and know what the results say. 

AUTHORS' RESPONSE: Unfortunatelly there were no significant differences in LA level (Fig. 1). On fig. 2 they are marked as follow:

significant difference

STH vs PAS

(p=0.008)

GRID vs PAS

(p=0.003)

How was VAS measured? Where participants asked to think of a certain area, was it general body pain? You need to include more details about the measurement of VAS. Also, do you have a VAS score prior to exercise? If assume individuals would report 0 on a VAS scale before exercise, but this is not true. Previous studies have found that participants can report anywhere from a 1-3 on the VAS scale at baseline. 

AUTHORS' RESPONSE: This issue has been adressed in methods section. Page 7, lines 158-160.

Page 8, Line 209: Pain was significantly reduced in all groups when?

 AUTHORS' RESPONSE: This has been clarified in the text.

Discussion 

Page 11, Lines 286-287: Your sentence starting off “Switching off by pressing…” is confusing. Consider revising for clarity. 

AUTHORS' RESPONSE: Included in the text.

You don’t discuss any limitations in your discussion. I’m sure your study had some and they should always be included.

AUTHORS' RESPONSE: Included in the text.

Tables 

Table 2 needs clarity in the legend. IAT, ART, and AFTER30 need to be defined as figures and tables should always stand alone. 

AUTHORS' RESPONSE: Included in the text. Legend has been clarified.

Also, you have no symbol for p<0.001 in the legend.

AUTHORS' RESPONSE: Indeed, we added that.

REVIEWER #2

Review for Plus one

Abstract: 

The abstract is written with many format/ writing issues: 1. The first paragraph is not a paragraph, and should be developed more that just one sentence. 2. Same for paragraph 2. 3. The overall writing is very confusing and at times does flow with any sense of organization. I suggest using something like intro, methods (Assessment, Timing and Treatments) then Results and Conclusion. 

AUTHORS' RESPONSE: Included in the text.

Introduction: 

Once again the first paragraph is not developed, and only has 2 sentences. Also the first paragraph leads me to think the paper is about lactate and Fatigue with no mention or hint of foam rolling. Paragraph 2 is also underdeveloped and needs to be expanded upon so that it discusses or sets up the paper for the topic of Foam rolling and DOMS.

AUTHORS' RESPONSE: Included in the text.

The use of IT in the paper specifically the intro is very confusing. I have a hard time following what “it” is in the many sentences. 

AUTHORS' RESPONSE: This has been included in the text as well as English grammar check has been provided by AJE. Please see reference # BFF0-200A-BB08-7F0E-1A7C

Line 73 please explain further with a reference or 2 to back of this claim., 

AUTHORS' RESPONSE: Included in the text.

Reduce the use of phrases like on the other hand in the intro

AUTHORS' RESPONSE: Included in the text.

Line 84, consider reorganizing to move the paragraph closer to the front of the intro.

AUTHORS' RESPONSE: As above we are writing about general effects of FR, so placing this here as more detailed seems to be justified. 

Finally please proof read the content as there are many writing and grammar mistakes in the introduction. 

AUTHORS' RESPONSE: Included in the text. Manuscript was edited for proper English language, grammar, punctuation, spelling, and overall style by AJE, reference # BFF0-200A-BB08-7F0E-1A7C

Methods: 

The methods section needs some rewriting as to develop more paragraphs that are not 1-2 sentences. 

AUTHORS' RESPONSE: Included in the text.

I am not sure about the 1 minute squat jumps, could you explain what reference or reasoning suggested this time and exercise?

AUTHORS' RESPONSE: During maximal efforts, the anaerobic (lactic) system lasts from 45 seconds to 2 minutes [Swanwick & Matthews, 2018]. That is why we choose 1 min. As we could observe in our previous studies [Adamczyk et al., 2014; Adamczyk et al., 2016] that this time along with maximal engagement of participants creates conditions close to volitional exhausted. Along with high LA concentration (mean for all participants 10.16 ±2.87) ranged 4.8-18.9 mmol/L warrant us that this exercise will be eligible for research aim i.e. “…inducing a significant increase in blood lactate levels as an effect glycolytic nature of exercise”.

Line 124: can you explain what you mean by adaption, is this meant to be a familiarization or some kind of temperature balance. 

AUTHORS' RESPONSE: Following explanation was added to the manuscript:

The purpose of the adaptation was to achieve a state of thermal balance in relation to the ambient temperature so that the obtained thermograms were reliable. Thanks to this, changes in the thermal image were the result of disturbances in production or heat dissipation as a result of the exercises [Ring & Ammer, 2012].

IS TSK the same as thermal imaging? They seem to be used interchangable at times. 

AUTHORS' RESPONSE: Tsk means skin temperature as a result of thermal image measurement. Appropriate explanation has been included in the text.

Also the same occurs with rolling, foam rolling or self myofascial release. Please clarify all these.

AUTHORS' RESPONSE: We used foam roller as a self-myofascial release technique and only in this context it has been used in the text.

I think it might be helpful in the writing to have less acronyms at times I had to keep going back to look at some of them that should be spelled out. ROI region of interest, or immediately after the test. 

AUTHORS' RESPONSE: To clarify this, we explained with every first use in each part of the manuscript.

Results:

Again I suggest removing the pronoun it and make the sentences stronger by stating what it is. 

There are several 1 sentence paragraphs that need to be developed or included in other sections of the results. 

AUTHORS' RESPONSE: Included in the text.

Discussion: 

Blood lactate changes: would like to see a few more comparisons of post research on lacate to the found results. And what significance does the reference on cardio bring to this paper?

AUTHORS' RESPONSE: Included in the text. Page 10, lines 253-255 

Please expand upon this section with more research refences to support or refute your findings. The massage paragraph needs to tie more into the relationship to foam rolling

AUTHORS' RESPONSE: Included in the text. Page 10, lines 259-263 / Page 11, lines 265-267 / Page 11, lines 273-274.

Line 254 when you say the results of research are you referring to your study or the literature. It is not clear. 

AUTHORS' RESPONSE: REFERENCE INCLUDED in the text. Page 11, line 277

Overall in the temperature section, I think you need to bring in your results as the compare contrast to the references you are using in this sections

Line 286: the sentence starting with switching is not a sentence please correct

AUTHORS' RESPONSE: Included in the text.

DOMS conclusion section: too many times “in light of “ is used also please expand your results to comparisons of the literature

AUTHORS' RESPONSE: Included in the text.

Overall I think the concept is interesting and I think the idea of trying to determine if foam rolling (specifically a type of roller) has an effect on preventing DOMS and improving recovery could be helpful to rehabilitation and sports performance. The methods performed were adequate, however the paper is not written in way that helps to clearly set up the research question. The introduction is littered with bad grammar and improper usage of terms like it and acronyms. The discussion is not a good comparison of the your results to the current literature and creates confusion as to what is out there versus what you found. 

I highly suggest some editing of this paper and or proof reading by someone else to catch the grammatical errors and typos. This would take a good project, which I liked reading about, more clear to the reader.

AUTHORS' RESPONSE: Included in the text. Manuscript was edited for proper English language, grammar, punctuation, spelling, and overall style by AJE, reference # BFF0-200A-BB08-7F0E-1A7C

---

## [Decision Letter · Decision Letter 1]

18 May 2020

PONE-D-20-05356R1

Does the type of foam roller influence the recovery rate, thermal response and DOMS prevention?

PLOS ONE

Dear Mr Adamczyk,

Thank you for submitting your manuscript to PLOS ONE. After careful consideration, we feel that it has merit but does not fully meet PLOS ONE’s publication criteria as it currently stands. Therefore, we invite you to submit a revised version of the manuscript that addresses the points raised during the review process.

ACADEMIC EDITOR: Myself, along with Reviewers' 1 and 2, thought that the manuscript was much improved with the edits of the resubmission. A member of PLOS ONE's Statistical Advisory Board was also asked to review the revised submission.  Please address the minor revision/comments of Reviewers 2 and 3, prior to acceptance.

We would appreciate receiving your revised manuscript by Jul 02 2020 11:59PM. To enhance the reproducibility of your results, we recommend that if applicable you deposit your laboratory protocols in protocols.io, where a protocol can be assigned its own identifier (DOI) such that it can be cited independently in the future. For instructions see: http://journals.plos.org/plosone/s/submission-guidelines#loc-laboratory-protocols

We look forward to receiving your revised manuscript.

Kind regards,

Kelly Naugle, PhD

Academic Editor

PLOS ONE

Reviewers' comments:

Reviewer's Responses to Questions

**Comments to the Author**

1. If the authors have adequately addressed your comments raised in a previous round of review and you feel that this manuscript is now acceptable for publication, you may indicate that here to bypass the “Comments to the Author” section, enter your conflict of interest statement in the “Confidential to Editor” section, and submit your "Accept" recommendation.

Reviewer #1: All comments have been addressed

Reviewer #2: All comments have been addressed

Reviewer #3: (No Response)

2. Is the manuscript technically sound, and do the data support the conclusions?

Reviewer #1: Yes

Reviewer #2: Yes

Reviewer #3: Partly

3. Has the statistical analysis been performed appropriately and rigorously? 

Reviewer #1: Yes

Reviewer #2: Yes

Reviewer #3: Yes

4. Have the authors made all data underlying the findings in their manuscript fully available?

Reviewer #1: Yes

Reviewer #2: Yes

Reviewer #3: Yes

5. Is the manuscript presented in an intelligible fashion and written in standard English?

Reviewer #1: Yes

Reviewer #2: Yes

Reviewer #3: Yes

6. Review Comments to the Author

Reviewer #1: Thank you for responding to my previous comments. I believe you have adequately answered all comments and the manuscript is ready for submission.

Reviewer #2: Much improvement on the writing and grammar. This was much easier to ready, thank you for the rewrites and the work put into editing.

Few minor notes:

Abstract paragraph 1: add foam into the correct spots

Still a few times that “it” is used and should be edited.

Line 93: expend upon how or why Hodgson questions long term affects

Line 139: what is the (REST acronym for) do you mean that REST is equal to adaptation and thermal images. If so that seems to be confusing.

Reference: please make sure the proper reference formats are being used throughout the entire reference sections.

Reviewer #3: In a randomized study of 33 males, pain levels were repeatedly assessed up to 96 hours post exercise using the Visual Analog Scale following foam rolling with smooth (STH), grid roller (GRID) or passive recovery (PAS). Additionally thermal imaging of skin temperature and blood lactate were repeatedly measured. In the PAS group, lactate concentration at 30 minutes was significantly lower than the other two groups. Pain scores decreased significantly in the STH group between 48 and 96 hours and systemic decreases in VAS scores were observed in the GRID group. No significant changes in VAS scores for the PAS group was observed.

Minor revisions:

1- In the abstract, indicate that the trial was randomized.

2- Move the sentences from lines 122-8 to the results section. Indicate the type of summary statistics provided in parenthesis.

3- Line 182: typographical error: Large effects had and eta-squared “>” 0.40.

4- Provide a summary table of the results corresponding to the details contained in lines 190-9. Include summary statistics, the overall p-value from the repeated measures ANOVA, and the pairwise Bonferroni corrected p-values (similar to Table 2).

5- Table 1: To conform to standard practice, transpose the matrix. List the groups in columns at the top and the biometric characteristics in the rows. Additionally, indicate that the summary values are means and standard deviations or standard errors of the means.

6- Table 2: Indicate the type of summary statistic following the +/- sign.

7- Be consistent with the notation for Tsk.

8- Page 21: Indicate if the confidence intervals shown on the graph were adjusted for repeated measures.

9- Recreate Figure 2 making it similar to Figure 1. Use a line plot and confidence intervals.

10- State and justify the study’s target sample size with a pre-study statistical power calculation. The power calculation should include: sample size, alpha level (indicating one or two-sided), minimal detectable difference and statistical testing method.

11- Indicate the date range subjects were enrolled in the study.

12- Clearly define the aims of the study.

7. PLOS authors have the option to publish the peer review history of their article (what does this mean?). If published, this will include your full peer review and any attached files.

Reviewer #1: No

Reviewer #2: No

Reviewer #3: No

---

## [Author Response · Author response to Decision Letter 1]

1 Jun 2020

Reviewer #2:

Much improvement on the writing and grammar. This was much easier to ready, thank you for the rewrites and the work put into editing. 

Few minor notes: 

Abstract paragraph 1: add foam into the correct spots

AUTHORS' RESPONSE: Included in lines 29 and 30.

Still a few times that “it” is used and should be edited. 

AUTHORS' RESPONSE: Included and edited in lines 70, 79, 335 and also “it” has been removed in lines 112, 249, 280. 

Line 93: expend upon how or why Hodgson questions long term affects

AUTHORS' RESPONSE: Included in lines 94-95.

Line 139: what is the (REST acronym for) do you mean that REST is equal to adaptation and thermal images. If so that seems to be confusing. 

AUTHORS' RESPONSE: This has been clarified in line 142, as REST is acronym of resting state before exercise.

Reference: please make sure the proper reference formats are being used throughout the entire reference sections. 

AUTHORS' RESPONSE: Included in the text.

Reviewer #3:

Minor revisions:

1- In the abstract, indicate that the trial was randomized.

AUTHORS' RESPONSE: Included in line 33.

2- Move the sentences from lines 122-8 to the results section. 

AUTHORS' RESPONSE: We were not sure about this remark. Are we supposed to do this? As lines 122-8 are description of participants so we believe it should stay in this part of the paper not in results section. Anyway we added “Participants” subheading in here.

Indicate the type of summary statistics provided in parenthesis.

AUTHORS' RESPONSE: Included in the text.

3- Line 182: typographical error: Large effects had and eta-squared “>” 0.40.

AUTHORS' RESPONSE: Included in the text.

4- Provide a summary table of the results corresponding to the details contained in lines 190-9. Include summary statistics, the overall p-value from the repeated measures ANOVA, and the pairwise Bonferroni corrected p-values (similar to Table 2).

AUTHORS' RESPONSE: Included in the text.

5- Table 1: To conform to standard practice, transpose the matrix. List the groups in columns at the top and the biometric characteristics in the rows. Additionally, indicate that the summary values are means and standard deviations or standard errors of the means.

AUTHORS' RESPONSE: Included in the text.

6- Table 2: Indicate the type of summary statistic following the +/- sign.

AUTHORS' RESPONSE: Included in the text.

7- Be consistent with the notation for Tsk.

AUTHORS' RESPONSE: We included that, one place without down script in line 243 has been corrected.

8- Page 21: Indicate if the confidence intervals shown on the graph were adjusted for repeated measures.

AUTHORS' RESPONSE: Included in the text.

9- Recreate Figure 2 making it similar to Figure 1. Use a line plot and confidence intervals.

AUTHORS' RESPONSE: Included in the text.

10- State and justify the study’s target sample size with a pre-study statistical power calculation. The power calculation should include: sample size, alpha level (indicating one or two-sided), minimal detectable difference and statistical testing method.

AUTHORS' RESPONSE: Following statement has been added „For the sample size (3 groups, each n = 11), assuming a typical significance level alpha = 0.05 (two-way test) and the standardized effect RMSSE = 0.77, the test power was 0.88” – lines 181-182.

11- Indicate the date range subjects were enrolled in the study.

AUTHORS' RESPONSE: The study was conducted in October 2019. Each group performed test separately but experimental sessions for groups, were held at the same time of the day. We’ve included this statement in the text, lines 127-128.

12- Clearly define the aims of the study.

Aims has been clarified in lines 118-119.

---

## [Editor Report · Decision Letter 2]

11 Jun 2020

Does the type of foam roller influence the recovery rate, thermal response and DOMS prevention?

PONE-D-20-05356R2

Dear Dr. Adamczyk,

We’re pleased to inform you that your manuscript has been judged scientifically suitable for publication and will be formally accepted for publication once it meets all outstanding technical requirements.

Kind regards,

Kelly Naugle, PhD

Academic Editor

PLOS ONE

Additional Editor Comments (optional):

Please fix the following grammatical errors in the manuscript:

Abstract, Line 33: Change "In randomized trial, enrolled..." to "This randomized trial enrolled.."

Page 4, line 79: change "In research is also suggested..." to "Research also suggests..."

Page 5, line 118: Change "Whether the type of roller in single foam rolling treatment influence on the..." to "whether the type of roller in single foam rolling treatment influences the rate...".
---

## [Editor Report · Acceptance letter]

18 Jun 2020

PONE-D-20-05356R2 

Does the type of foam roller influence the recovery rate, thermal response and DOMS prevention? 

Dear Dr. Adamczyk:

I'm pleased to inform you that your manuscript has been deemed suitable for publication in PLOS ONE. Congratulations! Your manuscript is now with our production department. 

Kind regards, 

on behalf of

Dr. Kelly Naugle 

Academic Editor

PLOS ONE